# Insights on the Biomarker Potential of Exosomal Non-Coding RNAs in Colorectal Cancer: An In Silico Characterization of Related Exosomal lncRNA/circRNA–miRNA–Target Axis

**DOI:** 10.3390/cells12071081

**Published:** 2023-04-04

**Authors:** Maria Mezher, Samira Abdallah, Ohanes Ashekyan, Ayman Al Shoukari, Hayat Choubassy, Abdallah Kurdi, Sally Temraz, Rihab Nasr

**Affiliations:** 1Department of Internal Medicine, Faculty of Medicine, American University of Beirut, Beirut 1107 2020, Lebanon; 2Department of Anatomy, Cell Biology and Physiological Sciences, Faculty of Medicine, American University of Beirut, Beirut 1107 2020, Lebanon; 3Department of Biochemistry and Molecular Genetics, Faculty of Medicine, American University of Beirut, Beirut 1107 2020, Lebanon; 4Department of Experimental Pathology, Immunology, and Microbiology, Faculty of Medicine, American University of Beirut, Beirut 1107 2020, Lebanon; 5Faculty of Sciences, Lebanese University, Beirut P.O. Box 6573, Lebanon; 6Department of Biochemistry and Molecular Genetics, American University of Beirut, Beirut 1107 2020, Lebanon

**Keywords:** colorectal cancer, non-coding RNAs, lncRNAs, circRNAs, miRNAs, exosomes, diagnosis, prognosis/prediction, biomarkers

## Abstract

Colorectal cancer (CRC) is one of the most common cancer types, ranking third after lung and breast cancers. As such, it demands special attention for better characterization, which may eventually result in the development of early detection strategies and preventive measures. Currently, components of bodily fluids, which may reflect various disease states, are being increasingly researched for their biomarker potential. One of these components is the circulating extracellular vesicles, namely, exosomes, which are demonstrated to carry various cargo. Of importance, the non-coding RNA cargo of circulating exosomes, especially long non-coding RNAs (lncRNAs), circular RNAs (circRNAs), and micro RNAs (miRNAs), may potentially serve as significant diagnostic and prognostic/predictive biomarkers. In this review, we present existing evidence on the diagnostic and prognostic/predictive biomarker value of exosomal non-coding RNAs in CRC. In addition, taking advantage of the miRNA sponging functionality of lncRNAs and circRNAs, we demonstrate an experimentally validated CRC exosomal non-coding RNA-regulated target gene axis benefiting from published miRNA sponging studies in CRC. Hence, we present a set of target genes and pathways downstream of the lncRNA/circRNA–miRNA–target axis along with associated significant Gene Ontology (GO) terms and Kyoto Encyclopedia of Genes and Genomes (KEGG) pathways, which may collectively serve to better characterize CRC and shed light on the significance of exosomal non-coding RNAs in CRC diagnosis and prognosis/prediction.

## 1. Introduction

Colorectal cancer (CRC) is among the most prevalent gastrointestinal tumors, accounting for an incidence of 10.2% worldwide. This makes it the third most common cancer after lung and breast cancers. Additionally, it is the second most common cause of cancer-related mortality [1]. Upward trends of CRC incidence have recently been observed, in part due to the aging population and the Western diet [2]. Although 2 to 5% of CRC cases can be attributed to hereditary familial disorders, such as Lynch syndrome, familial adenomatous polyposis (FAP), or Peutz–Jeghers syndrome, more than 90% of CRC cases remain sporadic [3,4]. The carcinogenesis process of CRC mostly occurs progressively over the span of several years. It originates from a premalignant adenomatous precursor, through the gradual accumulation of multiple mutations and epigenetic alterations, eventually transforming it into malignant carcinoma [5], hence the importance of screening methods and early detection of CRC to increase survival rates. Despite advancements in screening methods and the establishment of targeted treatments for this disease, a delayed diagnosis of progressive disease with localized or distant metastases is still responsible for the death of around 50% of CRC patients [6]. Furthermore, five-year survival rates of CRC patients can range from as high as 90% in early-stage disease to less than 13% for metastatic CRC [7]. Screening tests such as fecal occult blood, barium enema, sigmoidoscopy, and colonoscopy are currently available [8]. However, patient compliance with these tests has been mainly insufficient. This could be due to their inadequate sensitivity and specificity, high cost, and intrusive nature [6]. Moreover, tumor biomarkers in the blood, such as carbohydrate antigen 19-9 (CA 19-9) and carcinoembryonic antigen (CEA), are also often utilized clinically for CRC. Nonetheless, these biomarkers have limited sensitivity and specificity and thus cannot be used for screening or diagnosis of CRC on their own. They may accumulate in the blood as a result of benign conditions including pneumonia and inflammatory bowel disease as well as in smokers. Consequently, no non-invasive, accurate, and inexpensive screening techniques currently exist for the early detection of CRC [5].

Recently, a rising interest has been observed in circulating exosomal, non-coding RNAs as potential minimally invasive biomarkers for the diagnosis, prognosis, and treatment response of various cancer types, especially CRC [9]. Research has suggested that the aberrant expression of these molecules can be responsible for the regulation of complex cellular functions related to tumorigenesis and cancer progression [10].

Exosomes are the smallest among extracellular vesicles, ranging from 50 to 100 nm in diameter. They are composed of a phospholipid bilayer and are secreted by most cell types, including cancer cells, into all body fluids, including blood, plasma, urine, cerebrospinal fluid (CSF), and even amniotic fluid and saliva. They carry different types of molecules, including DNA, mRNA, non-coding RNA, proteins and lipids [9]. These exosomal cargoes’ strengths lie in the fact that they act as a stable molecular fingerprint of the cell of origin, making them suitable cancer biomarkers [11]. Exosomal vesicles play an important role in intracellular communication, cell signaling, immune response, cancer development and metastasis to specific organs [9]. Different types of exosomal RNAs are secreted by cancer cells into the tumor microenvironment, such as microRNAs, long non-coding RNAs and circular RNAs (Figure 1A). microRNAs (miRNAs, miRs) are endogenous, short, non-coding, single-stranded RNA sequences with an approximate length of 17 to 25 nucleotides. They have the ability to regulate gene expression post transcriptionally and control a range of biological processes such as cellular proliferation, differentiation, apoptosis, tumorigenesis, invasion and metastasis. A wide range of cancer pathological pathways, such as cancer development, classification, diagnosis and prognosis, are influenced by miRNA expression [12,13]. Long non-coding RNAs (lncRNAs) are another class of exosomal non-coding RNA with an estimated length exceeding 200 nucleotides [14]. A growing body of research has shown that lncRNAs are relatively stable and may have a crucial role in the onset and progression of cancer. Thus, they could be employed as accurate diagnostic and prognostic biomarkers [15]. Circular RNA (circRNA), a novel kind of endogenous non-coding RNA, has a covalent closed-loop structure. When released in exosomes, circRNAs have the ability to circulate through body fluids and initiate different cellular processes. Additionally, accumulating evidence has proven that circRNAs can be engaged in the pathophysiological functions of CRC cells, in particular by acting as miRNA molecular sponges [16]. Both lncRNAs and circRNAs are able to interact with miRNAs through the sponging effect and thus serve as competitors of endogenous RNAs (ceRNAs). microRNA response element (MRE) sequences, present within ceRNAs, allow for their binding to miRNAs (Figure 1B). This interaction blocks the role of miRNAs as inhibitors of mRNA translation into proteins. CeRNAs can thus control the quantity of the protein encoded by a certain gene or alter the epigenetic regulation of gene expression. This interplay of miRNAs and ceRNAs is particularly important in cancer development, because it has the ability to repress or promote carcinogenesis depending on the presence of tumor suppressor genes or oncogenes on the downstream end of the target miRNA [17,18,19,20].

Considering their involvement in multiple cancer pathological processes, their abundance in different bodily fluids, variety, and stable nature, exosomal non-coding RNAs make a perfect candidate for a prospective innovative liquid biopsy tool and have already been used in multiple malignancies, especially in CRC [11].

In this review, an in-depth approach will be used to explore the potential role of different types of exosomal non-coding RNAs in liquid biopsies for CRC diagnosis, prognosis, and therapy prediction, which has already been the focus of multiple studies in the literature. Finally, an in silico model will be used to develop an experimentally validated CRC exosomal non-coding RNA-regulated target gene axis by incorporating the miRNA sponging mode of action of lncRNAs and circRNAs. This model will help in the manipulation of this axis to develop new therapeutic strategies by providing better knowledge of the role of exosomal non-coding RNAs in CRC.

## 2. Search Strategy

The search strategy was centered around and based on three key chunks of our topic “Exosomal Non-Coding RNAs in Colorectal Cancer”, namely exosomes, non-coding RNAs (miRNAs, lncRNAs, and circRNAs), and colorectal cancer. The specific PubMed search input was the following: ((“Colorectal Neoplasms”(MeSH)) AND (“MicroRNAs”(MeSH)) OR “RNA, Long Noncoding”(MeSH)) OR (“RNA, Circular”(MeSH))) AND ((circulat*(Title/Abstract)) OR (“Exosomes”(MeSH))). We included the circulating component in our search and manually selected non-coding RNAs of exosomal origin only in an attempt to avoid missing any exosomal non-coding RNAs. All MeSH terms found below the utilized MeSH terms were incorporated as well. Other popular databases, such as MEDLINE Ovid and Embase, were also searched using a similar strategy, where no differences in the resulting article numbers were observed. Therefore, we relied on the PubMed hits for our review.

## 3. Exosomal Non-Coding RNAs as Potential Biomarkers in CRC

Exosomal non-coding RNAs (ncRNAs) may have a diagnostic, prognostic, or predictive potential or a mix of the aforementioned potentials.

### 3.1. Diagnostic Biomarker Potential of Exosomal ncRNAs in CRC

Exosomal ncRNAs play a key role in the control of tissue homeostasis and cell signaling, acting as a post-transcriptional mechanism of gene expression. The coordinated function of these molecules associated with other mechanisms avoids the development of aberrant cellular proliferation, regulates cellular differentiation and allows for the fine regulation of mRNAs. Many studies have demonstrated the role of exosomal ncRNAs including miRNAs, lncRNAs, and circRNAs as promising diagnostic biomarkers in CRC patients.

To start with miRNAs, their signatures appear to mirror pathological changes in CRC patients [7]. A study discovered a panel of seven serum exosomal miRNAs (let-7a, miR-1229, miR-1246, miR-150, miR-21, miR-223, and miR-23a) that could serve as highly sensitive future diagnostic biomarkers for CRC [21]. Their serum levels were notably higher in CRC patients compared to healthy controls, even in early disease, and their secretion was also increased in CRC tumor cells compared to normal colon cells [21]. Furthermore, Dokhanchi et al. [22] identified a panel of four diagnostic biomarkers for CRC: miR-19a-3p, miR-203-3p, miR-221-3p, and let-7f-5p, where they were significantly higher in CRC patients than in healthy individuals. In addition, CRC exosomal-derived miR-221-3p was shown to induce endothelial cell angiogenesis in vitro by targeting SOCS3 [22]. miR-92b serves as a diagnostic biomarker for CRC, where it was shown to be significantly downregulated in CRC patients when compared to those with colorectal adenoma (AD) and other non-cancerous (NC) lesions [23]. Moreover, it was significantly decreased with high-grade intraepithelial neoplasia patients than in patients with NC lesions [23]. Furthermore, Li et al. found that miR-96-5p and miR-149 are downregulated in tissue and plasma from CRC patients and in GPC1+ exosomes in plasma from CRC patients compared to healthy controls [24]. It has been shown that two months after surgical intervention, these levels were normalized. Moreover, the overexpression of these miRNAs significantly increased cell death, while it decreased cell proliferation as well as GPC1 expression in CRC cell lines, plasma of mice bearing these cell line tumors, and xenograft tumors [24]. In addition, Wang et al. showed that expression levels of miR-10a were significantly lower in serum and cancer tissue samples from patients with CRC compared to that of healthy individuals and paired non-cancerous tissues [25]. It has been shown that the proliferative and migratory activities of primary normal human lung fibroblasts (NHLFs) and the expression levels of IL-6, IL-8 and IL-1β in NHLFs were reduced by CRC cell exosomal-derived miR-10a [25]. Furthermore, exosomal miR-377-3p and miR-381-3p were shown to be downregulated in CRC patients and early-stage CRC compared to healthy controls [26]. It was shown by ROC curve analysis that they may serve as diagnostic biomarkers for CRC, including in the early stage [26]. Moreover, seven overexpressed exosomal miRNAs in colorectal adenoma (CRA) organoids: miR-4323, miR-4284, miR-1268a, miR-1290, miR-6766-3p, miR-21-5p, and miR-1246, were determined by Handa et al. [27]. These miRNAs could be potential diagnostic biomarkers to CRA according to ROC analysis [27].

Second, as mentioned earlier, lncRNAs are transcripts of more than 200 nucleotides in length that regulate gene expression at various levels [7]. lncRNA CCAT2 was overexpressed in the serum of CRC patients when compared to healthy controls. It was shown that the higher expression of this lncRNA was associated with local invasion and lymph node metastasis [28]. Moreover, six potential lncRNAs in circulatory exosomes (LNCV6_116109, LNCV6_98390, LNCV6_38772, LNCV_108266, LNCV6_84003, and LNCV6_98602) were significantly upregulated in the plasma of CRC patients compared to normal individuals, which makes them potential diagnostic biomarkers in colorectal cancer [29].

### 3.2. Diagnostic and Prognostic Biomarker Potential of Exosomal ncRNAs in CRC

miR-874 was shown to be downregulated in CRC patients compared to benign adenoma (AD) cases and healthy controls [30]. It was shown to have the potential of discriminating CRC patients from healthy controls and patients with AD by ROC analysis [31]. In addition, its low serum level was associated with advanced stages, positive lymph node metastasis, and distant metastasis [30]. Moreover, it was identified as an independent prognostic factor for OS of CRC patients. Collectively, miR-874 is a diagnostic and prognostic biomarker for CRC [30]. Other miRNAs that were found to be downregulated in CRC patients as compared to healthy controls are miR-4461 and miR-150-5p [31,32]. It has been shown that BMSCs-derived exosome miR-4461 downregulated COPB2 expression and inhibited HCT116 and SW480 cells migration and invasion [31]. It has been shown that the level of miR-150-5p was greatly increased one-month post-surgical treatment in post-operative blood samples. Moreover, decreased levels of this miRNA were correlated with poor differentiation, advanced stages, poor OS, and poor DFS [32]. In addition, the combination of miR-150-5p with CEA was shown to be the best at increasing the diagnostic accuracy of CRC from healthy controls [33]. Furthermore, it was determined that patients with higher levels of miR-150-5p have longer survival times than those with lower expression [32]. Collectively, serum exosomal miR-4461 and miR-150-5p have diagnostic and prognostic potentials [31,32]. Lastly, Zhang et al. discovered a positive feedback loop leading to CRC progression where miR-17-5p was shown to be upregulated in CAFs exosomes and is secreted from CAFs to CRC cells via the RUNX3/MYC/TGF-β1 signaling pathway to promote CRC progression and metastasis [33].

In contrast, miR-6803-5p was significantly increased in serum samples from patients with CRC compared to healthy controls, which was associated with advanced TNM stages as liver metastasis, poor OS, and poor DFS [34]. In addition, it was shown that miR-6803-5p was associated with poor prognosis in CRC by Cox regression analysis [34]. Wang et al. showed that miR-125a and miR-320c serve as diagnostic and prognostic biomarkers, where they are upregulated in the plasma of CRC patients, especially in early-stage CRC [35]. Moreover, it has been shown that adding miR-125a to CEA results in a better predictive model than that of CEA alone in differentiating CRC patients from HCs [36]. Another finding by Jiang et al. indicated that exosomal miR-200c-3p suppresses CRC migration and invasion and increases their death [36]. It has been proposed as a potential therapeutic target for CRC and as a diagnostic marker [36]. Dai et al. [37] showed that circulating exosomal miR-424-5p promotes CRC proliferation and metastasis [37].

LncRNA RPPH1 was shown to be overexpressed in CRC tissues and exosomes compared to healthy controls [38]. This high expression was associated with advanced TNM stages, poor overall survival (OS), poor disease-free survival (DFS) and poor prognosis [38]. The interaction of RPPH1 with β-III tubulin (TUBB3) averts its ubiquitination and induces EMT [38]. In addition, RPPH1 induces metastasis and proliferation by being transported into macrophages, leading to their M2 polarization [38]. Collectively, RPPH1 serves as a diagnostic and prognostic biomarker [38]. Moreover, Oehme et al. found that HOTTIP was shown to be downregulated in CRC serum exosomes compared to healthy controls as well as being an independent prognostic marker for OS, which mirrors its diagnostic and prognostic potentials [39]. In addition, lncRNA ADAMTS9-AS1, which was shown to be downregulated in CRC in comparison to healthy controls, has been demonstrated to play an important role in regulating CRC tumorigenesis through β-catenin suppression [40]. Li et al. [40] also identified that this lncRNA may have potential clinical applications in CRC prognosis and treatment as an ideal therapeutic target [40]. CRNDE-h was shown, on the other hand, to be upregulated in CRC cell lines and serum exosomes compared to benign colorectal disease (BCD) and healthy donors, in a study conducted by Liu et al. [41]. Moreover, the levels of this lncRNA were correlated with regional lymph node metastasis, distant metastasis, and low OS [41]. In addition, an inverse relation was determined between GAS5 and miR-221 in CRC tissues, serum and exosomes in a study by Liu et al., where the downregulation of GAS5 was associated with an upregulation of miR-221 [42]. These levels correlated with CRC stage, lymph node metastasis, local recurrence rate, and distant metastasis rate [42].

Furthermore, circLPAR1 was identified as a promising predictor in CRC diagnosis, as it showed cancer specificity in diagnosing CRC [43]. circLPAR1 had an improved diagnostic performance in comparison to CEA and CA 19-9 [43]. In addition, its downregulation in CRC tissues was associated with increased OS [43].

Table 1 outlines the current literature evidence of these exosomal ncRNAs as potential diagnostic biomarkers in CRC.

### 3.3. Prognostic/Predictive Biomarker Potential of Exosomal ncRNAs in CRC

Several studies have shed light on the prognostic/predictive potential of exosomal ncRNAs by promoting metastasis of CRC. A plethora of circulating exosomal ncRNAs has been found to have the potential to serve as colorectal cancer prognostic biomarkers (Table 2). As such, miRNAs such as miR-934 and miR-25-3p were shown to be possible promoters of colorectal cancer liver metastasis via M2 polarization and promoting vascular permeability and angiogenesis, respectively [44,45]. Moreover, Teng et al. perceived that significantly higher levels of exosomal miR-193a-5p are found in the exosomes of metastatic CRC cell lines and plasma of CRC patients with more advanced illness, such as liver metastasis [46]. Cell cycle assessment revealed the role of miR-193a-5p in causing G1 arrest and repressing cell proliferation by targeting Caprin1 [46]. Moreover, a study by Cheng et al. showed that miR-146a-5p, the major miRNA in colorectal cancer stem cells (CRCSC), endorses tumorigenicity and stem-like characteristics in recipient CRC cells via targeting the Numb gene [47]. High serum levels of exosomal miR-146a-5p display more CRCSCs with miR-146a^High^/Numb^Low^ traits, an elevated level of tumor-infiltrating CD66(+) neutrophils and a reduced number of tumor-infiltrating CD8(+) T cells [47]. Moreover, CRC exosome-derived miR-221/222 was shown to activate liver hepatocyte growth factor (HGF) by targeting SPINT1 expression by in vitro studies [48]. Importantly, through in vivo studies, miR-221/222 was shown to play an important role in forming a favorable premetastatic niche (PMN) that leads to the aggressive nature of CRC [48]. In addition, in vivo and in vitro studies by Liu et al. showed that high metastatic CRC-delivered exosomal miR-106b-3p promoted cell migration, invasion, epithelial-to-mesenchymal transition (EMT), and lung metastasis [49]. Furthermore, hypoxia-induced exosomal miR-135a-5p was correlated with the development, clinical severity, and prognosis of CRC liver metastases through the premetastatic niche [50]. Exosomal miR-135a-5p initiated the large tumor suppressor kinase 2-yes-associated protein-matrix metalloproteinase 7 axis to promote the occurrence of CRC liver metastasis [50]. miR-27b-3p secreted by EMT-CRC cells increased blood vessel permeability and facilitated the generation of circulating tumor cells (CTC) [51]. miR-27b-3p attenuated the expression of vascular endothelial cadherin (VE-Cad) and p120 at the post-transcriptional level by binding to the 3′-untranslated region of VE-Cad and p120 directly [51]. Clinically, the plasma exosomal miR-27b-3p levels were positively correlated with malignant progression and CTC count in CRC patients [51]. Yang et al. were able to prove clinically that miR-106b-5p was significantly elevated in CRC tissues and negatively correlated with the levels of PDCD4 in CRC specimens, and high expression of exosomal miR-106b-5p in plasma was significantly associated with the malignant progression of CRC [52]. The increased level of miR-106b-5p activated the (PI3K)γ/AKT/mammalian target of the rapamycin (mTOR) signaling cascade by directly suppressing programmed cell death 4 (PDCD4) in a post-transcription level, which contributed to the M2 polarization of macrophages [52]. In turn, activated M2 macrophages, in a positive-feedback manner, promoted EMT-mediated migration, invasion, and metastasis of CRC cells [52]. Furthermore, serum exosomal miR-6869-5p is a promising circulating biomarker for the prediction of CRC prognosis [53]. miR-6869-5p was shown to inhibit cell proliferation and production of the inflammatory cytokines (TNF-α and IL-6) in CRC cells via directly targeting TLR4 [53]. miR-6869-5p functions as a tumor suppressor through its protective effect that is dependent on the TLR4/NF-κB signaling pathway [53]. In addition, the 3-year survival rate was poor among CRC patients with decreased levels of miR-6869-5p in serum exosomes, which demonstrates its important role in disease prognosis [53].

On another note, circFMN2 was discovered by Li et al. [54] to have a markedly increased expression level in CRC tissues, cell lines and exosomes from CRC patient serum. It has been shown that circFMN2 knockdown drastically reduced in vitro cell migration and proliferation [54]. It was also linked to progression of CRC via an miR-1182/hTERT axis [54]. Moreover, circ-133 was shown to be enriched in the plasma exosomes of CRC patients and to increase with disease progression [55]. Exosomal circ-133 derived from hypoxic cells was delivered into normoxic cells and promoted cancer metastasis by acting on the miR-133a/GEF-H1/RhoA axis [55]. In addition, it has been shown that the knockdown of circ-133 can inhibit tumor metastasis [55]. These data reveal that exosomal circ-133 is expected to be a new biomarker for monitoring tumor progression and might be a novel therapeutic target [55]. Lastly, the research by Shang et al. was the first to demonstrate that CRC-derived exosomal circPACRGL, from CRC patient serum, played an oncogenic role in CRC cell proliferation, invasion and migration, and in differentiation of N1 to N2 neutrophils via the miR-142-3p/miR-506-3p-TGF-*β1* axis [56].

Some studies have shed light on the important roles of exosomal ncRNAs in promoting chemoresistance or chemosensitivity of CRC cells to anticancer drugs. Ning et al. [57] found that miR-208b was upregulated in oxaliplatin-resistant CRC cells and in the serum of FOLFOX-resistant CRC patients. Their study showed that CRC-secreted exosomal miR-208b targeted PDCD4, thus enhancing the expansion of Treg cells and oxaliplatin resistance [57].

Moreover, the sponging effect of exosomal lncRNA NNT-AS1 on miR-496 was investigated and was found to contribute to their potential roles as prognostic biomarkers for CRC [14].

Hon et al. [58] studied hsa-circ_0000338 in FOLFOX (5-Fu and oxaliplatin)-resistant CRC patients. They found that it was significantly upregulated in non-responders’ serum compared to that of responders [58]. Moreover, the knockdown of exosomal hsa-circ-0000338 improved the chemoresistance of CRC cells [58]. They showed that hsa-circ-0000338 was selectively transferred from HCT116-R (resistant) to HCT116-P (parental) cells upon co-culture, where the latter demonstrated increased viability against drug treatment in comparison to control cells [58]. Moreover, serum exosomal circRNA hsa_circ_0005963 was shown to stimulate glycolysis to develop chemoresistance through the miR-122-PKM2 axis [59]. This intercellular signal transmission raises the possibility of a brand-new therapeutic target and lays the groundwork for possible clinical uses in drug-resistant CRC in the future [59].

On the other hand, Baek et al. showed the potential of exosomal ncRNAs to promote chemosensitivity in CRC cells. They determined that upregulated levels of serum exosomal miR-199b-5p correlated with better chemoradiotherapy (CRT) response [60].

**Table 2 cells-12-01081-t002:** Prognostic/predictive biomarker potential of exosomal ncRNAs in CRC.

Non-Coding RNA(s)	Non-Coding RNA Class	Number of CRC Patients	Source	Biomarker Potential	Status	Clinical Evidence	References
miR-208b	miRNA	N/A	Serum	Predictive	Upregulated in non-responders compared to responders	Yes, resistance to FOLFOX therapy in CRC patients	[57]
miR-934	MiRNA	N/A	Serum	Prognostic	Upregulated in stage IV CRC compared to stage I CRC	Yes, promotes CRLM by regulating the crosstalk between CRC cells and TAMs	[44]
miR-25-3p	MiRNA	N/A	Serum	Prognostic	Upregulated in metastatic CRC tissues compared to those without metastasis	Yes, involved in pre-metastatic niche formation	[45]
miR-135a-5p	MiRNA	N/A	Serum	Prognostic	Upregulated in CRC tissues post metastasis compared to those premetastasis	Yes, correlates with the development, clinical severity, and prognosis of CRC liver metastases through the premetastatic niche	[50]
Circ-133	CircRNA	N/A	Plasma	Prognostic	Upregulated in advanced CRC stages compared to early stage	Yes, promotes CRC metastasis by acting on miR-133a/GEF-H1/RhoA axis	[55]
miR-27b-3p	MiRNA	N/A	Serum	Prognostic	Upregulated in metastatic CRC patients compared to non-metastatic	Yes, promotes metastasis, increases blood vessel permeability and facilitate the generation of CTCs	[51]
miR-106b-5p	MiRNA	N/A	Plasma	Prognostic	Upregulated in CRC patients compared to normal ones	Yes, mediates CRC invasion, migration and metastasis	[52]
miR-6869-5p	miRNA	N/A	Serum	Prognostic	Downregulated in advanced CRC patients	Yes, inhibits cell proliferation and the production of inflammatory cytokines	[53]
RPPH1	LncRNA	52	Plasma	Prognostic	Upregulated in treatment-naïve CRC patients compared to post tumor resection	Yes, promotes CRC metastasis	[38]
HOTTIP	LncRNA	100	Serum	Prognostic	Downregulated in CRC patients compared to healthy donors	Yes, associated with worse prognosis and poor overall survival	[39]
NNT-AS1	LncRNA	40	Serum	Prognostic	Upregulated in preoperative compared to postoperative CRC samples and in advanced stage compared to early stage CRC patients	Yes, promotes CRC proliferation, migration, and invasion in CRC patients	[14]
circ0000338	CircRNA	17	Serum	Predictive	Upregulated in chemoresistant CRC patients compared to chemosensitive	Yes, confers chemoresistance in CRC patients	[58]
hsa-circ-0005100 (circFMN2)	CircRNA	35	Serum	Prognostic	Upregulated in CRC patients compared to healthy ones	Yes, promotes tumor proliferation in CRC patients	[54]
miR-200c	miRNA	N/A	Serum	Prognostic/Predictive	Upregulated in advanced compared to early stage CRC patients	No	[61]
miR-6803-5p	miRNA	168	Serum	Prognostic	Upregulated in CRC patients compared to healthy controls	Yes, correlated with poor prognosis and low OS in CRC patients	[34]
miR-193a-5p	miRNA	25	Plasma	Prognostic	Upregulated in advanced CRC stage compared to early stage	Yes, promotes tumor progression in CRC patients	[46]
miR-146a-5p	MiRNA	53	Serum	Prognostic	Upregulated in CRC patients compared to normal	Yes, promotes tumor stemness expansion	[47]
miR-150-5p	MiRNA	133	Serum	Prognostic	Downregulated in preoperative CRC patient samples compared to postoperative ones	Yes, low levels correlate with poor prognosis, worse OS, and shorter survival	[32]
miR-199a/b-3p, miR-199a-5p and miR-199b-5p	MiRNA	89	Serum	Prognostic and Predictive	Downregulated in advanced CRC stage compared to early stage	Yes, upregulated levels are associated with superior RFS and OS	[60]
miR-874	MiRNA	125	Serum	Prognostic	Downregulated in CRC patients compared to benign AD	Yes, low levels correlate with positive distant metastasis, lymph node metastasis, poor differentiation and advanced TNM stage	[30]
miR-221/222	MiRNA	N/A	Serum	Prognostic	Upregulated in metastatic CRC patient samples compared to those without metastasis	Yes, promotes liver metastasis in CRC patients	[48]
miR-106b-3p	miRNA	80	Serum	Prognostic	Upregulated in patients with metastasis compared to those without metastasis	Yes, promotes CRC metastasis and high serum level correlate with poor prognosis	[49]
circPACRGL	circRNA	N/A	Serum	Prognostic	Upregulated in CRC patients	Yes, promotes CRC progression and metastasis	[56]
circ_0005963	circRNA	19	Serum	Prognostic	Upregulated in oxaliplatin resistant patients compared to oxaliplatin sensitive patients	Yes, promotes glycolysis and oxaliplatin resistance	[59]

## 4. In Silico Analysis of CRC Exosomal lncRNA/circRNA–miRNA–Target Axis

### 4.1. Rationale

We sought to identify shared target genes and pathways downstream of the CRC exosomal ncRNAs axis using an in silico approach as described in Ashekyan et al. [17] based primarily on miRTargetlink 2.0 [62]. The utilized approach takes advantage of the sponging mechanism of action of lncRNAs and circRNAs on miRNAs, which in turn leads to relief in the inhibition exerted by the respective miRNA on the target mRNA, leading to the upregulation of the target gene. We consider that uncovering the genes and/or pathways downstream of this axis reveals and better characterizes the diagnostic, or prognostic/predictive biomarker potential of exosomal ncRNAs in CRC. The input of our analysis was strictly literature based, where we manually selected publications in which the miRNA sponging mechanism of action of exosomal lncRNAs and circRNAs was apparent. We selected hsa-miR-577, sponged by circ_0007334 [63], hsa-miR-17-5p, sponged by circLONP2 [64], hsa-miR-766-5p, sponged by circ_0094343 [16], hsa-miR-1182, sponged by circFMN2 [54], hsa-miR-141, sponged by lncRNA H15 [65], hsa-miR-142-3p, sponged by circPACRGL [56], hsa-miR-122, sponged by circ_005963 [59], hsa-miR-26a/b, sponged by lncRNA MALAT1 [66], hsa-miR-133a, sponged by circ 133 [55], hsa-miR-496, sponged by lncRNA NNT-AS1 [14], hsa-miR-217 and hsa-miR-485-3p, sponged by circ_0000338 [67], and hsa-miR-343, sponged by LINC00659 [68], all of which are presented in Figure 2 as the colorectal cancer exosomal lncRNA/circRNA–miRNA–target axis.

### 4.2. Shared Target Genes and Pathways

Using the list of miRNAs sponged by either lncRNAs or circRNAs (Figure 2), the miRTargetLink 2.0 tool returned a set of 38 strong experimentally validated target genes downstream of the CRC exosomal axis shared between at least two miRNAs from the literature set (Figure 3). The literature review revealed that the vast majority of the identified genes had previously been linked to colorectal cancer pathogenesis. Of these genes, the ones that seem to have an important role in CRC are EZH2, PTEN, EGFR, IGF1, HGF, SMAD4 and PTGS2 [69,70,71,72,73,74,75,76,77,78,79,80,81,82,83,84,85,86,87,88,89,90,91,92,93,94,95,96,97,98,99,100,101,102,103,104].

Using the same set of exosomal ncRNA-sponged miRNAs, miRTargetLink 2.0 also returned a large set of strong and weak experimentally validated pathways shared between at least three miRNAs from the literature set. Of special importance and implication in CRC, a few shared pathways, such as the P53, PI3K-AkT, TGF-β, and Wnt signaling pathways, are depicted in Figure 4 [105,106,107,108].

### 4.3. Gene Ontology and KEGG Pathways

The DAVID functional annotation and enrichment analysis tool (2021 update) [109] was used to more deeply characterize the functional implications of the 38 shared target genes downstream of the CRC exosomal ncRNAs axis output by miRTargetLink as CRC diagnostic, and prognostic/predictive biomarkers. To this end, Gene Ontology (GO) terms enrichment [110] and Kyoto Encyclopedia of Genes and Genomes (KEGG) pathway [111] analyses were conducted. As such, the GO terms enrichment analysis is presented in Table 3, where significant biological process, molecular function, and cellular component GO terms are shown.

In addition, KEGG pathway analysis of the resultant gene set (Table 4) revealed a multitude of significantly associated pathways, where differing numbers of these genes were found to play a role. Several pathways implicated in CRC, such as “Pathways in cancer”, “PI3K-Akt signaling pathway”, “MAPK signaling pathway”, and “P53 signaling pathway”, were found to be among these significant pathways, where the target genes downstream of the CRC exosomal ncRNAs axis were found to be implicated with differing frequencies.

Lastly, in an attempt to further characterize the functional implications of the exosomal lncRNA/circRNA–miRNA–target axis in CRC, we used our manually curated set of lncRNA/circRNA-sponged miRNAs from the literature as an input to the miRNA functional enrichment analysis and annotation tool (miEAA 2.0)’s over-representation analysis (ORA) [112]. MiEAA 2.0 allows for the investigation of functional enrichment of a set of miRNAs with selectable emphasis options, such as target genes, pathways, diseases, Gene Ontology, annotations, and localization, among others. Of interest, we selected the miRNA pathway enrichment option, where this information is brought via miRWalk 2.0 [113], a comprehensive atlas of predicted and experimentally verified microRNA-target gene/pathway interactions. It specifically offers a framework to obtain statistically significant miRNA interactions on genes associated with pathways. With a *p* value significance threshold of 0.025 to obtain the most significant pathways and a minimum required hits per sub-category, i.e., minimum number of miRNAs per pathway equaling two along with FDR (Benjamini–Hochberg) *p* value adjustment, the 15 inputted miRNAs were found to be over-represented in 32 pathways with differing frequencies and significance values (Figure 5). Among the resultant pathways, a considerable number of CRC-related signaling pathways were apparent among which TGF-β signaling, MAPK cascade, and PI3 kinase signaling complement and confirm our preceding miRTargetLink 2.0 and KEGG pathway analyses findings. In addition, a few signature pathways are also demonstrated as over-represented, which may eventually contribute to CRC progression. Examples are hypoxia response by HIF activation, hedgehog signaling, G-protein signaling pathway, Ras pathway, platelet-derived growth factor (PDGF) signaling, and epidermal growth factor (EGF) receptor signaling (Figure 5).

## 5. Discussion

Exosomal ncRNAs have been the subject of increasing research due to their potential role as stable, non-invasive biomarkers for the diagnosis, prognosis, and therapeutic response of many cancer types, particularly CRC. In this review, we highlighted multiple sets of exosomal ncRNAs and classified them depending on their prospective diagnostic or prognostic/predictive capacities in colorectal cancer, through a thorough literature review. Additionally, by adding the miRNA sponging mechanism of action of these lncRNAs and circRNAs, an in silico model was established, and an experimentally validated colorectal cancer exosomal ncRNAs-regulated target gene axis was created.

Subsequently, we established that ncRNAs can serve as both diagnostic and prognostic/predictive biomarkers. In our review, sixteen ncRNAs were found to have a dual role as diagnostic and prognostic/predictive biomarkers in colorectal cancer. These are exosomal miR-874 [30], miR-4461 [31], miR-150-5p [32], miR-17-5p [33], miR-6803-5p [34], miR-125a and miR-320c [35], miR-200c-3p [36], miR-424-5p [37], RPPH1 [38], HOTTIP [39], ADAMTS9-AS1 [40], CRNDE-h [41], GAS5 [42], and circLPAR1 [43].

On another note, in silico analysis produced a set of 38 strong experimentally validated target genes downstream of the CRC exosomal axis, the majority of which were implicated in CRC pathogenesis. Furthermore, KEGG pathway analysis of this gene set demonstrated the significant involvement of a large number of these genes in multiple pathways. Most importantly, we found that 16 of the 38 target genes played a role in “Pathways in cancer” (Appendix A). This finding was supported by the literature, where these genes were also found to be associated with cancer pathways, of which we mention EGFR, COX-2, PTEN, IGF, HGF, and SMAD4 [81,82,83,86,91,92,93,94,95,96,97,98,99,100,114,115,116,117,118,119,120,121,122,123,124]. Additionally, we established the implication of 10 of the 38 target genes in the PI3K-Akt signaling pathway, one of the essential pathways of colorectal cancer pathogenesis (Appendix A) [106,125,126,127].

### 5.1. Advantages of Exosome-Derived Biomarkers

Tumor-derived exosomes carry various molecules, such as proteins and miRNA profiles derived from their parental cell, and thus provide an accurate depiction of the features of the tumor cells from which they originated. Numerous studies have shown a strong correlation between exosomal biomarkers and tissue analyses [7]. Furthermore, the presence of a lipid bilayer membrane encapsulating exosomes makes them extremely stable and preserves them against degradation while in circulation. This makes it possible to perform various distinct downstream analyses from these vesicles alone after the collection and isolation of exosomes [128]. Moreover, the relative stability of these vesicles at ambient temperature decreases the risk of potential variability with specimen acquisition and storage. This along with their excellent specificity makes them perfect biomarkers for early cancer diagnosis [128]. It is crucial for diagnostic biomarkers to be highly specific to type of cancer, in order to reduce the rates of false positives and to improve their diagnostic value for clinical use [129]. On another note, analysis of the RNA transcriptome of the tumor can be conducted through exosomal RNA. Other advantages of exosomal biomarkers include their abundant prevalence in body fluids and the possibility of long-term storage of exosomal blood samples in a biobank [130,131].

### 5.2. Limitations and Challenges of Exosome-Derived Biomarkers

Several limitations were identified. First, many studies that were used in this review relied on small sample sizes [22,27,31,38,46,47,54,58]. Another limitation is that the studies that were included had targeted different stages of CRC, ranging from adenoma [27] and early-stage CRC [23,29,35] to advanced metastatic CRC [45]. While this variability reflects the inclusivity of the studies for all stages of the disease, it limits the possibility of validating any results. Future studies on exosomal ncRNA in same-stage CRC should be conducted in order to withdraw valid and universal results. The same issue was also noted for the type of biomarker or ncRNA in question. We observed a large heterogeneity in the literature in terms of the exosomal ncRNAs being studied in CRC. There was a lack of studies validating the same exosomal ncRNAs in CRC. This weakens the validity of the corresponding ncRNAs, especially for circRNAs and lncRNAs. This variability was also observed in terms of the source of the exosomes being studied. Some researchers used serum-derived exosomes while others used plasma-derived exosomal ncRNAs. This also affects the validity of the results.

On the other hand, there are numerous obstacles regarding the use of exosomal ncRNAs as CRC biomarkers. The main challenge in this regard remains as technical difficulties with exosome isolation. The most effective method for isolating exosomes is ultracentrifugation, which requires a great amount of time and physical effort, necessitates many starting materials, is expensive in terms of equipment, and is ineffective for high-throughput tests. Moreover, particle clumping from ultracentrifugation prevents the identification of individual vesicles. Commercial kits for isolating exosomes are readily available, but they lack standardization and produce low-purity results. Additionally, since whole particle extraction is not feasible, they use buffers that degrade exosome membranes, which restrict qualitative and quantitative studies. Furthermore, due to their small size and composition, it is challenging to create a gold standard for quantification [131,132]. The quantification of biomarkers is one of the major issues with their use. This problem has been well documented for circulating miRNAs and must be resolved in order to reliably identify and measure biomarkers in clinical samples [129].

Finally, one of the main challenges for researchers aiming to establish novel exosomal diagnostic biomarkers is the necessity to adhere to the Standards for Reporting of Diagnostic Accuracy (STARD), which ensure the precision and completeness of study reporting [133].

## 6. Conclusions

Exosomal ncRNA, such as lncRNAs, circRNAs and miRNAs, were found to play a critical role in CRC oncogenesis and therapy resistance. Exosomes isolated from blood samples have shown diagnostic potential in the early stages of the illness, prognostic potential to the TNM stage as well as overall survival (OS) and disease-free survival (DFS), and predictive potential to chemotherapy-resistant CRC, specifically oxaliplatin. Furthermore, it has been demonstrated that a combination of exosomal ncRNAs with CEA levels in the blood can enhance the diagnostic and predictive accuracy of CEA. Despite the technical difficulties restricting their clinical implementation, the data gathered are encouraging, and additional research is needed to improve our understanding of exosomes and their potential value as cancer biomarkers. Furthermore, a target gene axis for CRC exosomal ncRNAs was developed from experimentally validated data. The downstream 38 hallmark genes of this axis were identified, and literature evidence supported the implication of these genes in different levels of colorectal cancer pathogenesis. Further evidence implicating this target gene axis in CRC-related pathways has validated our hypothesis. This target gene collection could inspire future research to better elucidate the different mechanisms of CRC pathophysiology.

## Figures and Tables

**Figure 1 cells-12-01081-f001:**
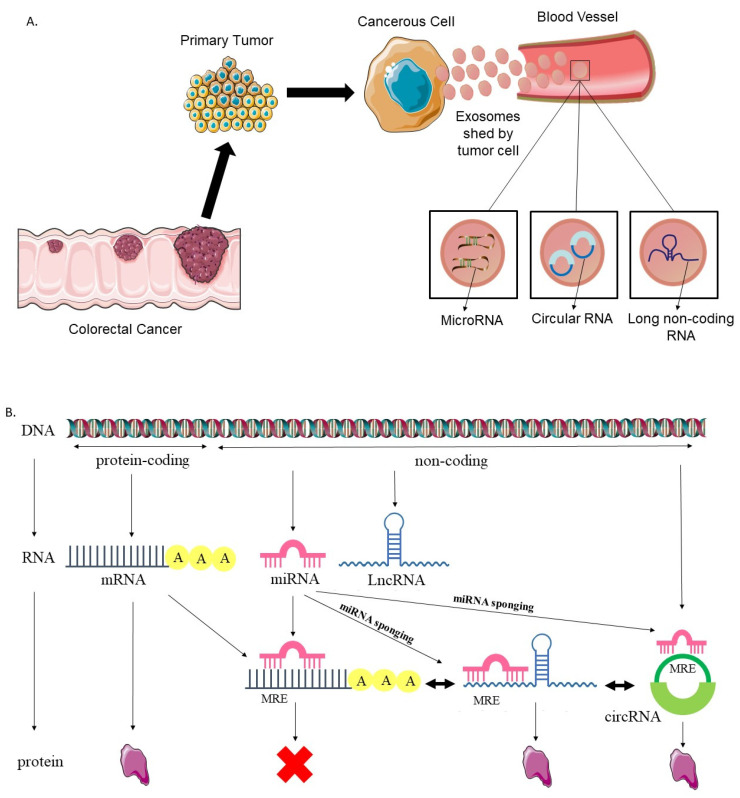
(**A**) Exosomal microRNA, circular RNA, and long non-coding RNAs in liquid biopsies have potential roles for CRC diagnosis, prognosis, and therapy prediction. (**B**) The rationale behind this sponging mechanism lies in the fact that mRNAs, which are produced from protein-coding sections of the genome, are normally translated into proteins. On the other hand, miRNAs, lncRNAs, and circRNAs are transcribed from non-coding DNA. miRNAs are capable of binding to mRNAs by MREs, which can thus prevent them from being translated. Since they also contain MREs, lncRNAs and circRNAs may bind to and sponge miRNAs, causing the previously miRNA-inhibited mRNAs to be translated into proteins.

**Figure 2 cells-12-01081-f002:**
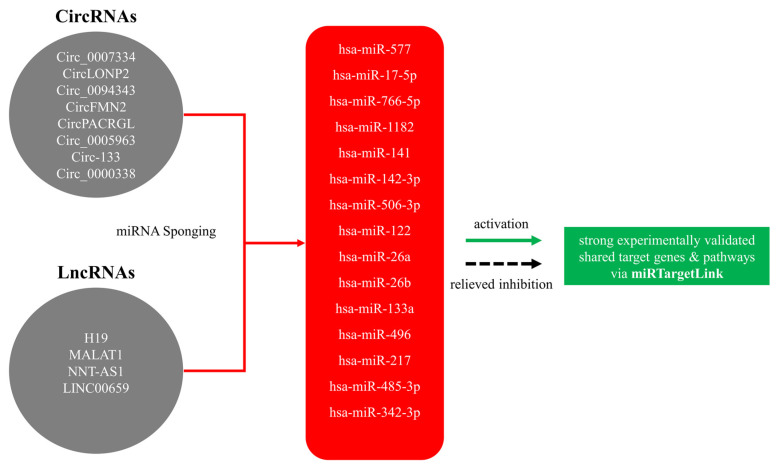
The colorectal cancer exosomal lncRNA/circRNA–miRNA–target axis.

**Figure 3 cells-12-01081-f003:**
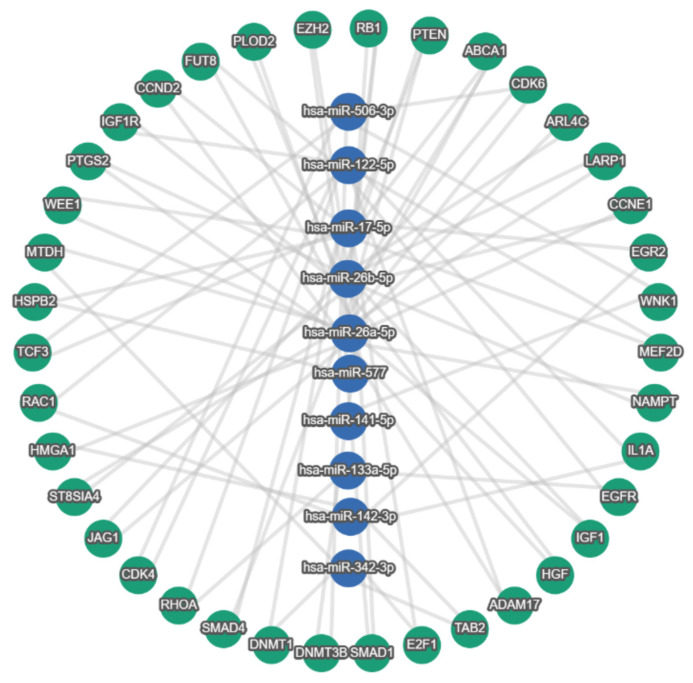
Strong experimentally validated shared target genes downstream of the CRC exosomal lncRNA/circRNA–miRNA–target axis. Green nodes indicate genes, and blue nodes indicate miRNAs. Minimum miRNA threshold = 2.

**Figure 4 cells-12-01081-f004:**
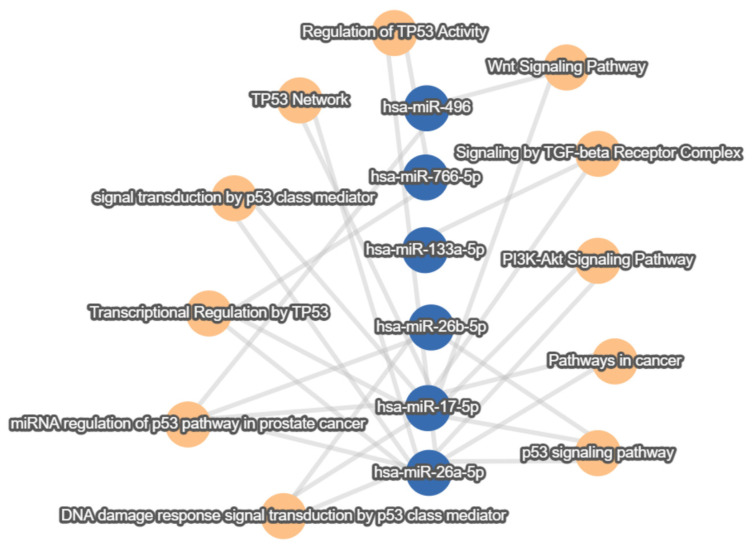
Strong and weak experimentally validated shared target pathways downstream of the CRC exosomal lncRNA/circRNA–miRNA–target axis. Orange nodes indicate pathways, and blue nodes indicate miRNAs. Minimum miRNA threshold = 3.

**Figure 5 cells-12-01081-f005:**
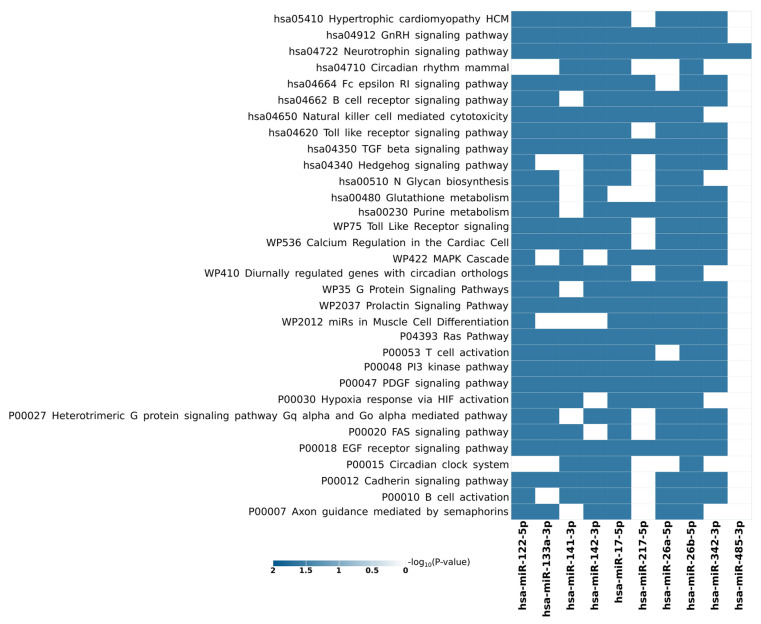
Exosomal lncRNA/circRNA-sponged miRNAs functional enrichment analysis via miEAA. Thirty-two pathways were found to be significantly enriched by our list of 15 literature-curated miRNAs in the exosomal lncRNA/circRNA–miRNA–target axis. *p* value threshold = 0.025. FDR (Benjamini–Hochberg) *p* value adjustment. Minimum required hits per subcategory = 2.

**Table 1 cells-12-01081-t001:** Diagnostic biomarker potential of exosomal ncRNAs in CRC.

Non-Coding RNA(s)	Non-Coding RNA Class	Number of CRC Patients	Source	Biomarker Potential	Status	Clinical Evidence	References
circLPAR1	circRNA	112	Plasma	Diagnostic	Downregulated	Yes	[43]
Let-7a, miR-1229, miR-1246, miR-150, miR-21, miR-223, and miR-23a	miRNA	99	Serum	Diagnostic	Upregulated	Yes	[21]
CCAT2	LncRNA	100	Serum	Diagnostic	Upregulated	Yes	[28]
RPPH1	LncRNA	52	Plasma	Diagnostic	Upregulated	Yes	[38]
HOTTIP	LncRNA	100	Serum	Diagnostic	Downregulated	Yes	[39]
miR-92b	miRNA	40	Plasma	Diagnostic	Downregulated	Yes	[23]
miR-6803-5p	miRNA	168	Serum	Diagnostic	Upregulated	Yes	[34]
miR-96-5p and miR-149	miRNA	102	Plasma	Diagnostic	Downregulated	Yes	[24]
miR-125a, miR-320c	miRNA	50	Plasma	Diagnostic	Upregulated	Yes	[35]
MiR-150-5p	miRNA	133	Serum	Diagnostic	Downregulated	Yes	[34]
MiR-4461	miRNA	15	BMSC	Diagnostic	Downregulated	Yes	[31]
MiR-10A	miRNA	40	Primary CRC cells	Diagnostic	Downregulated	Yes	[25]
MiR-874	miRNA	125	Serum	Diagnostic	Downregulated	Yes	[30]
MiR-19a-3p, miR-203-3p, miR-221-3p, let-7f-5p	miRNA	36	Serum	Diagnostic	Upregulated	Yes	[22]
MiR-377-3p, miR-381-3p	miRNA	175	Serum	Diagnostic	Downregulated	Yes	[26]
LNCV6_116109, LNCV6_98390, LNCV6_38772, LNCV_108266, LNCV6_84003, and LNCV6_98602	LncRNA	50	Plasma	Diagnostic	Upregulated	Yes	[29]
CRNDE-h	LncRNA	148	Serum	Diagnostic	Upregulated	Yes	[41]
GAS5	LncRNA	158	Serum	Diagnostic	Downregulated	Yes	[42]
MiR-17-5p	miRNA	N/A	Serum	Diagnostic	Upregulated	No	[33]
ADAMTS9-AS1	LncRNA	886	Serum	Diagnostic	Downregulated	Yes	[40]
miR-4323, miR-4284, miR-1268a, miR-1290, miR-6766-3p, miR-21-5p, and miR-1246	miRNA	26	Serum	Diagnostic	Upregulated	Yes	[27]
miR-200c-3p	miRNA	N/A	Serum	Diagnostic	Downregulated	No	[36]
miR-424-5p	miRNA	N/A	Serum	Diagnostic	Upregulated	No	[37]

**Table 3 cells-12-01081-t003:** Gene Ontology (GO) terms enrichment analysis of the shared target genes downstream of the CRC exosomal ncRNAs axis. Minimum gene count = 5. FDR cut-off = 0.05.

**Biological Process**		
**GO Term**	**Gene Count**	**FDR**
positive regulation of transcription from RNA polymerase II promoter	14	1.30 × 10^−4^
negative regulation of transcription from RNA polymerase II promoter	10	7.80 × 10^−3^
positive regulation of cell proliferation	9	1.80 × 10^−3^
regulation of transcription from RNA polymerase II promoter	9	2.00 × 10^−1^
positive regulation of gene expression	8	6.30 × 10^−3^
positive regulation of transcription, DNA-templated	8	2.80 × 10^−2^
negative regulation of cell proliferation	7	1.80 × 10^−2^
negative regulation of apoptotic process	7	2.80 × 10^−2^
regulation of transcription, DNA-templated	7	1.80 × 10^−1^
signal transduction	7	3.30 × 10^−1^
G1/S transition of mitotic cell cycle	6	1.30 × 10^−4^
response to hypoxia	6	3.70 × 10^−3^
positive regulation of cell migration	6	1.30 × 10^−2^
response to drug	6	1.80 × 10^−2^
cell division	6	3.70 × 10^−2^
positive regulation of fibroblast proliferation	5	1.30 × 10^−3^
positive regulation of smooth muscle cell proliferation	5	1.80 × 10^−3^
Aging	5	2.90 × 10^−2^
positive regulation of protein phosphorylation	5	3.50 × 10^−2^
regulation of gene expression	5	5.10 × 10^−2^
regulation of cell cycle	5	9.70 × 10^−2^
nervous system development	5	1.50 × 10^−1^
protein phosphorylation	5	2.00 × 10^−1^
negative regulation of transcription, DNA-templated	5	3.00 × 10^−1^
cell differentiation	5	3.80 × 10^−1^
**Molecular Function**		
**GO Term**	**Gene Count**	**FDR**
protein binding	36	3.40 × 10^−3^
protein kinase binding	9	1.00 × 10^−3^
RNA polymerase II core promoter proximal region sequence-specific DNA binding	8	1.10 × 10^−1^
identical protein binding	8	3.00 × 10^−1^
chromatin binding	7	1.30 × 10^−2^
DNA binding	7	3.00 × 10^−1^
RNA polymerase II sequence-specific DNA binding transcription factor binding	6	3.20 × 10^−3^
enzyme binding	6	3.20 × 10^−2^
transcription factor activity, sequence-specific DNA binding	6	9.50 × 10^−2^
kinase activity	5	3.20 × 10^−2^
transcriptional activator activity, RNA polymerase II transcription regulatory region sequence-specific binding	5	1.60 × 10^−1^
**Cellular Component**		
**GO Term**	**Gene Count**	**FDR**
Nucleus	24	1.30 × 10^−3^
Cytoplasm	23	1.30 × 10^−3^
Cytosol	19	4.50 × 10^−2^
Nucleoplasm	18	3.80 × 10^−3^
Chromatin	11	1.30 × 10^−3^
Membrane	10	1.60 × 10^−1^
intracellular membrane-bounded organelle	7	5.70 × 10^−2^
endoplasmic reticulum membrane	7	6.60 × 10^−2^
transcription factor complex	6	1.30 × 10^−3^
macromolecular complex	6	5.70 × 10^−2^
apical plasma membrane	5	5.00 × 10^−2^
focal adhesion	5	5.70 × 10^−2^
Golgi membrane	5	1.50 × 10^−1^

**Table 4 cells-12-01081-t004:** KEGG pathway implications of the shared target genes downstream of the CRC exosomal ncRNAs axis. Minimum gene count = 5. FDR cut-off = 0.05.

KEGG Pathway	Gene Count	FDR
Pathways in cancer	16	1.00 × 10^−8^
MicroRNAs in cancer	11	2.90 × 10^−6^
Human papillomavirus infection	10	2.10 × 10^−5^
PI3K-Akt signaling pathway	10	3.40 × 10^−5^
Melanoma	9	1.00 × 10^−8^
Breast cancer	9	1.50 × 10^−6^
Hepatocellular carcinoma	9	2.90 × 10^−6^
Human T-cell leukemia virus 1 infection	9	1.40 × 10^−5^
Glioma	8	3.70 × 10^−7^
Cell cycle	8	5.20 × 10^−6^
Cellular senescence	8	1.40 × 10^−5^
Focal adhesion	8	5.20 × 10^−5^
Epstein-Barr virus infection	8	5.20 × 10^−5^
Viral carcinogenesis	8	5.30 × 10^−5^
Human cytomegalovirus infection	8	9.60 × 10^−5^
Pancreatic cancer	7	5.20 × 10^−6^
Small cell lung cancer	7	1.40 × 10^−5^
Prostate cancer	7	1.40 × 10^−5^
Endocrine resistance	7	1.40 × 10^−5^
Proteoglycans in cancer	7	5.60 × 10^−4^
MAPK signaling pathway	7	2.70 × 10^−3^
Non-small cell lung cancer	6	5.00 × 10^−5^
p53 signaling pathway	6	5.10 × 10^−5^
FoxO signaling pathway	6	6.40 × 10^−4^
Measles	6	7.70 × 10^−4^
Gastric cancer	6	9.60 × 10^−4^
Cushing syndrome	6	1.10 × 10^−3^
Hepatitis B	6	1.30 × 10^−3^
Transcriptional misregulation in cancer	6	2.70 × 10^−3^
Kaposi sarcoma-associated herpesvirus infection	6	2.70 × 10^−3^
Rap1 signaling pathway	6	3.70 × 10^−3^
Ras signaling pathway	6	6.00 × 10^−3^
Adherens junction	5	7.00 × 10^−4^
Chronic myeloid leukemia	5	8.50 × 10^−4^
EGFR tyrosine kinase inhibitor resistance	5	9.50 × 10^−4^
Signaling pathways regulating pluripotency of stem cells	5	6.50 × 10^−3^
Hepatitis C	5	8.90 × 10^−3^

## Data Availability

Not applicable.

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
