# Peer review of "Insights on the Biomarker Potential of Exosomal Non-Coding RNAs in Colorectal Cancer: An In Silico Characterization of Related Exosomal lncRNA/circRNA–miRNA–Target Axis"

_cells, 2023, doi:10.3390/cells12071081_

Round 1

Reviewer 1 Report

The manuscript entitled “Insights on the Biomarker Potential of Exosomal Non-Coding RNAs in Colorectal Cancer: An in-Silico Characterization of Related Exosomal lncRNA/circRNA-miRNA-Target Axis” represents a narrative revision of the literature, conducted by Mezher et al.

Although several authors have previously and recently approached and reviewed the topic (doi.org/10.1186/s12943-021-01318-6; doi:10.2147/CMAR.S166308,

doi.org/10.3892/ol.2021.12863, doi.org/10.1017/erm.2022.21, etc…), the present manuscript addressed the potential diagnostic and prognostic biomarker role of ncRNAs rather that the biogenesis and mechanistic aspects, adding a new reading key to the literature.

However, in my opinion, the work is not well organized, getting difficult in following to the readers.

Regarding the schematization of the work, I have some difficulties understanding which is the difference between subsections 3.1, 3.2 and 3.3. I suggest considering just two subsections: “Diagnostic biomarker potential of ncRNAs in CRC” and “Prognostic/Predictive biomarker potential of ncRNAs in CRC”. In each subgraph miRNAs, lncRNAs and circRNAs should be consistently discussed. For instance, some miRNAs' diagnostic role is wrongly presented in 3.2 subsection (i.e. line 196).

Importantly, in order to protect them from degradation, miRNAs released by normal and cancer cells are bound to proteins, such as Argonaute, or lipoproteins or packed in exosomes. It is estimated that only 10% of circulating miRNAs is transported in the extracellular vesicles, including exosomes (DOI: 10.1016/j.cmet.2019.07.011). Therefore, I suggest changing the title and the content of the manuscript, referring to the “circulating” ncRNAs. On the other hand, i.e.  Radwan et al. (reference 21) in their study does not refer to “exosomal miRNAs” but to “plasma miRNAs”, and in the methods nothing was reported about the exosome extraction and isolation. In conclusion, the authors should always refer to “circulating miRNAs” and when appropriate, specify “exosomal miRNAs”

In table 2, I suggest specifying which is the observed clinical evidence. I.e.: for miR-208 upregulation: resistance to FOLFOX therapy in CRC patients.

Minor comments

After using for the first time the acronyms ncRNAs (line 135), miRNAs (line 82), lncRNAs (line 88) and circRNAs (line 92), please be sure to use them, when necessary, throughout the text (i.e. line 137, line 185, line 234, line 318, etc..).

Line 225: please clarify that “RPPH1” is a lncRNA.

Author Response

1- Regarding the schematization of the work, I have some difficulties understanding which is the difference between subsections 3.1, 3.2, and 3.3. I suggest considering just two subsections: “Diagnostic biomarker potential of ncRNAs in CRC” and “Prognostic/Predictive biomarker potential of ncRNAs in CRC”. In each subgraph miRNAs, lncRNAs and circRNAs should be consistently discussed. For instance, some miRNAs' diagnostic role is wrongly presented in 3.2 subsection (i.e. line 196).

We thank the reviewer for this feedback and understand this point. Through our literature search, we found that several ncRNAs (16 in our review) (such as miR-874, miR-4461, RPPH, circLPAR1…) have dual function or potentials in CRC, both diagnostic and prognostic, and thus fall under the two titles (diagnostic + prognostic/predictive). So, in order to prevent redundancy and duplication in the text, we have considered three subsections:

3.1. ncRNAs with pure diagnostic potential

3.2. ncRNAs with both diagnostic and prognostic potentials

3.3. ncRNAs with pure prognostic/predictive potential

2- Importantly, in order to protect them from degradation, miRNAs released by normal and cancer cells are bound to proteins, such as Argonaute, or lipoproteins or packed in exosomes. It is estimated that only 10% of circulating miRNAs is transported in the extracellular vesicles, including exosomes (DOI: 10.1016/j.cmet.2019.07.011). Therefore, I suggest changing the title and the content of the manuscript, referring to the “circulating” ncRNAs. On the other hand, i.e.  Radwan et al. (reference 21) in their study does not refer to “exosomal miRNAs” but to “plasma miRNAs”, and in the methods nothing was reported about the exosome extraction and isolation. In conclusion, the authors should always refer to “circulating miRNAs” and when appropriate, specify “exosomal miRNAs” 

We thank the reviewer for highlighting this mistake. We now deleted this reference and kept only those that are detected only in exosomes.

As for the choice of exosomal ncRNAs versus circulating free ncRNAs, we appreciate the point of the reviewer. However, although exosomal ncRNAs account for a small percentage, they have been shown in literature to be essential for cell-cell communication in the cancer microenvironment. The proliferation of cancer cells, angiogenesis, metastasis, treatment resistance, and immunomodulation are all processes in which exosomal ncRNAs play a significant role as regulators in the development of cancer. Moreover, it was shown by many studies that exosomes reflect the property of cancer cells and greatly enhance sensitivity and specificity by enabling the detection of biomarkers versus total serum and other circulating biomarkers (Ogata-Kawata H, et al. Circulating exosomal microRNAs as biomarkers of colon cancer. PLoS ONE. 2014).

3- In table 2, I suggest specifying which is the observed clinical evidence. I.e.: for miR-208 upregulation: resistance to FOLFOX therapy in CRC patients.

We thank the reviewer for this important suggestion. This is now addressed in the revised manuscript.

4- Minor comments

  • After using for the first time the acronyms ncRNAs (line 135), miRNAs (line 82), lncRNAs (line 88) and circRNAs (line 92), please be sure to use them, when necessary, throughout the text (i.e. line 137, line 185, line 234, line 318, etc..).
  • -Line 225: please clarify that “RPPH1” is a lncRNA.

We thank the reviewer for their comments. These comments have been addressed and modified accordingly in the revised manuscript.

Reviewer 2 Report

This manuscript presents a well-written and thorough review of the diagnostic and prognostic value of non-coding RNAs in colorectal cancer. As CRC is one of the most frequent cancers worldwide and shows an increasing trend, there is an urgent need to identify new biomarkers for this cancer, so this review is justified. The literature reviewed is extensive and up-to-date. The second part where the authors develop their in-silico model based on the sponging mechanism is an original approach that adds value to the review. There are some points that I would like to discuss:

1.   Why do the authors limit their review to the ncRNAs contained in exosomes? As they discuss later in the review, exosomes are hard to translate to the clinic due to limitations in their isolation, purification, and quantification. Would it be easier to take into account all extracellular vesicles? In this case, for instance, magnetic beads can be used to isolate them based on the CD63 marker or similar and perhaps it could be easily implemented in a clinical setup.

2.       I think the sponging mechanism could be explained more clearly on pages 2-3, lines 95-105, and maybe even add an explanatory figure to facilitate the understanding of the mechanism to the audience.

3.       In Table 2, where the authors specify the prognostic value of ncRNA, it is not clear what is meant by upregulated/downregulated. Are ncRNA upregulated in CRC, are upregulated in advanced/metastatic patients, are upregulated in chemoresistant patients, etc.?

4.       In the 3.3. section it is not clear enough whether some of the miRNAs the authors present are clinically validated or not, as it seems they are only reporting their function but not whether they have been isolated from patients and tested for their validity. I’m mostly referring to miR-934, miR-25-3p, miR-221/222, miR-106b-3p, miR-27b-3p, and miR-106b-5p.

5.       It would be nice if the authors could add a section discussing the underlying mechanisms or pathways affected by the ncRNA identified. Some of them are already discussed in the text (cell cycle progression, MYC/TGFb pathway, inflammatory pathways, etc.), but can the ncRNA be grouped into different pathways affecting CRC development, progression, or resistance to treatment? I think this point could greatly improve the impact of the review.

6.       With all the papers analyzed, has any of them considered sex/gender analysis? Could there be differences in these markers for sex/gender? The same with ethnicity. Are these papers reporting their results based on a single, homogenous population? I think this could be another limitation to add to the 5.2 section, as we tend to overgeneralize results and then they are not reproducible enough.

7.       In the conclusion section, lines 495-496, the authors state the following: “Furthermore, a target gene axis for CRC exosomal non-coding RNAs was developed and experimentally validated.” With this wording, it seems that they conducted experiments to validate this ncRNA. I think it would be more correct to say “... was developed from experimentally validated data”.

8.       Minor point: it seems that on page 4, the first paragraph of the 3.1 section (lines 138-143) is missing a reference.

9.       Minor point: on page 8, lines 282 and 285 are referring to CTC without having defined before the abbreviation. I think it means circulating tumor cells, but for some audiences it may not be clear, please specify.

10.   Minor point: on page 9, line 322, it is not necessary to define PDCD4 as it has already been done before.

11.   Minor point: the authors refer to Supplementary Figure 1 that I could not access.

Author Response

1- Why do the authors limit their review to the ncRNAs contained in exosomes? As they discuss later in the review, exosomes are hard to translate to the clinic due to limitations in their isolation, purification, and quantification. Would it be easier to take into account all extracellular vesicles? In this case, for instance, magnetic beads can be used to isolate them based on the CD63 marker or similar and perhaps it could be easily implemented in a clinical setup.

We appreciate the comment of the reviewer. Although exosomal non coding RNAs research is still in its infancy and have some limitations, they have been shown in literature to be essential for cell-cell communication in the cancer microenvironment. The proliferation of cancer cells, angiogenesis, metastasis, treatment resistance, and immunomodulation are all processes in which exosomal ncRNAs play a significant role as regulators in the development of cancer. Moreover, it was shown by many studies that exosomes reflect the property of cancer cells and greatly enhance sensitivity and specificity by enabling the detection of biomarkers versus total serum and other circulating biomarkers (Ogata-Kawata H, et al. Circulating exosomal microRNAs as biomarkers of colon cancer. PLoS ONE. 2014).

2- I think the sponging mechanism could be explained more clearly on pages 2-3, lines 95-105, and maybe even add an explanatory figure to facilitate the understanding of the mechanism to the audience.

We thank the reviewer for their comment. A new figure 1B illustrating the sponging mechanism with a detailed legend has been added to the revised manuscript.

The following description was added to the legend to better explain the sponging mechanism “The rationale behind this sponging mechanism lies in the fact that mRNAs, which are produced from protein-coding sections of the genome, are normally translated into proteins. On the other hand, miRNAs, lncRNAs, and circRNAs are transcribed from the non-coding DNA. miRNAs are capable of binding to mRNAs by MREs and can thus prevent them from being translated. Since they also contain MREs, lncRNAs and circRNAs, may bind to and sponge miRNAs, causing the previously miRNA-inhibited mRNAs to be translated into proteins.”

3- In Table 2, where the authors specify the prognostic value of ncRNA, it is not clear what is meant by upregulated/downregulated. Are ncRNA upregulated in CRC, are upregulated in advanced/metastatic patients, are upregulated in chemoresistant patients, etc.?

We thank the reviewer for this important suggestion. This is now addressed in the revised manuscript.

4- In the 3.3. section it is not clear enough whether some of the miRNAs the authors present are clinically validated or not, as it seems they are only reporting their function but not whether they have been isolated from patients and tested for their validity. I’m mostly referring to miR-934, miR-25-3p, miR-221/222, miR-106b-3p, miR-27b-3p, and miR-106b-5p.

We appreciate the comment of the reviewer. miR-934, miR-25-3p, miR-221/222, miR-106b-3p, miR-27b-3p, and miR-106b-5 were all detected in serum/plasma samples collected from CRC patients as mentioned in the table.

5- It would be nice if the authors could add a section discussing the underlying mechanisms or pathways affected by the ncRNA identified. Some of them are already discussed in the text (cell cycle progression, MYC/TGFb pathway, inflammatory pathways, etc.), but can the ncRNA be grouped into different pathways affecting CRC development, progression, or resistance to treatment? I think this point could greatly improve the impact of the review.

We appreciate the suggestion of the reviewer. In order to address the reviewer’s recommendation and better characterize the identified ncRNAs’ functional axis in CRC, we added an additional small in-silico-based functional enrichment analysis via the microRNA enrichment analysis and annotation tool (miEAA) while concentrating on the underlying target pathways. Please see lines (424-446) and figure 5 in the revised manuscript. Since we present the lncRNA/circRNA-miRNA-target axis as a potential axis for targeting CRC development/progression/resistance to treatment, we have attempted to further functionally characterize the lncRNA/circRNA-sponged miRNAs in CRC and hence our new analysis and output, which interestingly confirms and complements our initial findings regarding implicated target pathways.

6- With all the papers analyzed, has any of them considered sex/gender analysis? Could there be differences in these markers for sex/gender? The same with ethnicity. Are these papers reporting their results based on a single, homogenous population? I think this could be another limitation to add to the 5.2 section, as we tend to overgeneralize results and then they are not reproducible enough.

We thank the reviewer for highlighting this important point. Kindly note that gender and ethnicity differences in regard to ncRNAs expression were not studied in the articles we cited. However, regarding ethnicity, the following statement was added to the limitations section of the manuscript: "Furthermore, it is important to highlight that ncRNAs can differ between different ethnic groups which also limits the possibility of generalizing the results (135)."

7- In the conclusion section, lines 495-496, the authors state the following: “Furthermore, a target gene axis for CRC exosomal non-coding RNAs was developed and experimentally validated.” With this wording, it seems that they conducted experiments to validate this ncRNA. I think it would be more correct to say “... was developed from experimentally validated data”.

We thank the reviewer for their comment. This comment has been addressed and modified accordingly in the manuscript.

 8- Minor point: it seems that on page 4, the first paragraph of the 3.1 section (lines 138-143) is missing a reference.

We thank the reviewer for their comment. We would like to clarify that the paragraph at the beginning of section 3.1 (lines 138-143) is an introductory paragraph that was originally written by one of the authors and was not taken from any reference.

9- Minor point: on page 8, lines 282 and 285 are referring to CTC without having defined before the abbreviation. I think it means circulating tumor cells, but for some audiences it may not be clear, please specify.

We thank the reviewer for their comment. This comment has been addressed and modified accordingly in the manuscript.

10- Minor point: on page 9, line 322, it is not necessary to define PDCD4 as it has already been done before.

We thank the reviewer for their comment. This comment has been addressed and modified accordingly in the manuscript.

11- Minor point: the authors refer to Supplementary Figure 1 that I could not access.

We thank the reviewer for their comment. We have now added supplementary figures at the end of the revised manuscript

Reviewer 3 Report

The review is well written and cover all the main arguments the others planed to discuss. The right and relevant  papers are discussed and cited.

Author Response

We would like to thank you for the time and efforts taken in reviewing our manuscript and for the positive comment. 

Round 2

Reviewer 1 Report

The authors have clearly improved the manuscript and made it really clearer. However, the in silico analysis needs to be revised by a bioinformatic or computational biotechnologist.

Author Response

As per the reviewer’s comment we asked Mr. Abdallah Kurdi, the bioinformatician at our faculty to review the in silico analysis and this is feedback:

“I checked the "in-silico" methodology and it's all good: the methodology and the tools are used correctly. I just found two typos here: line 422 "exoso-mal" should be "exosomal" and line 443 "Raspathway" should be "Ras pathway". Everything is OK!”

Now we corrected the above mistakes and added his name as coauthor on this review